# IGLOO: Machine Vision System for Determination of Solubilization Index in Phosphate-Solubilizing Bacteria

**DOI:** 10.3390/microorganisms13040860

**Published:** 2025-04-09

**Authors:** Pablo José Menjívar, Andrés Felipe Solis Pino, Julio Eduardo Mejía Manzano, Efrén Venancio Ramos Cabrera

**Affiliations:** 1Facultad de Ingeniería, Corporación Universitaria Comfacauca—Unicomfacauca, Cl. 4 N. 8-30, Popayán 190001, Cauca, Colombia; pablomenjivar@unicomfacauca.edu.co; 2Escuela de Ciencias Básicas, Tecnología e Ingeniería—ECBTI, Universidad Nacional Abierta y a Distancia—UNAD, Calle 5 # 46N-67, Popayán 190001, Cauca, Colombia; julioe.mejia@unad.edu.co; 3Escuela de Ciencias Agrícolas, Pecuarias y del Medio Ambiente—ECAPMA, Universidad Nacional Abierta y a Distancia—UNAD, Calle 14 # 28-45, Pasto 520001, Nariño, Colombia; efren.ramos@unad.edu.co

**Keywords:** machine vision, phosphate solubilization, YOLOv8, solubilization index, image segmentation

## Abstract

Phosphorus is an important macronutrient for plant development, but its bioavailability in soil is often limited. Phosphate-solubilizing microorganisms play a vital role in phosphorus biogeochemistry, offering a sustainable alternative to chemical fertilizers, which pose environmental risks. Manual measurements for quantifying phosphate solubilization capacity are laborious, subjective, and time-consuming, so there is a need to develop more efficient and objective approaches. This study aimed to develop and validate a machine vision system called IGLOO to automate and optimize the determination of relative phosphate solubilization efficiency in phosphate-solubilizing bacteria. IGLOO was developed using YOLOv8 in conjunction with creating and labeling a dataset of images of bacterial colonies grown in vitro with the bacterial strains Enterobacter R11 and FCRK4. The model was trained with a different number of epochs. IGLOO’s performance was evaluated by comparing its segmentation accuracy with accepted metrics in the domain and by contrasting its solubilization efficiency estimates with experts’ manual measurements. The model achieved greater than 90% accuracy for colony and halo detection, with a relative error of less than 6% compared to manual measurements, demonstrating its reliability by minimizing observer variability. Finally, IGLOO represents a significant advance in the quantitative evaluation of phosphate solubilization of microorganisms because it reduces analysis time and provides objective and reproducible results for agricultural studies.

## 1. Introduction

Phosphorus (P) is an essential macronutrient for plant development, whose availability in the soil is limited by its predominance in ionic forms of low solubility [1]. Soil microorganisms, including fungi and bacteria, can solubilize inorganic phosphate and organic phosphate, playing a fundamental role in the biogeochemistry of P in the rhizosphere and the surrounding soil. Phosphorus-solubilizing microorganisms enhance soil P availability to plants, representing up to 40% of the soil bacterial population [2]. The presence of phosphorus-solubilizing microorganisms in the soil increases the availability of this essential nutrient for plants, which produce enzymes such as phytases, acid and alkaline phosphatases, and phosphodiesterases, which hydrolyze organic phosphorus compounds, releasing soluble inorganic phosphate that plants can absorb. This process is fundamental to the physiological functioning of plants, as phosphorus is crucial in photosynthesis, energy transfer, and nucleic acid synthesis [3].

An essential process performed by these microorganisms is phosphorus assimilation, which is directly linked to plant growth. Chemically synthesized fertilizers are commonly used to meet phosphorus needs in productive agricultural systems. However, their application often results in significant losses due to runoff, leaching, and retention in soil colloids [1]. This phosphorus deficiency can be mitigated by using phosphate-solubilizing microorganisms, such as certain fungi and bacteria, which increase phosphorus availability, which is crucial for plant nutrition [4]. Among the main phosphorus-solubilizing microorganisms are Pseudomonas, Bacillus, and Aspergillus [5]. In addition, Enterobacter has been shown to be effective in promoting plant growth by solubilizing phosphorus and producing phytohormones, which can enhance nutrient uptake [6].

When characterizing the phosphorus solubilization of a microorganism in a solid medium, two groups can be identified, such as late solubilizers and non-solubilizers, which are determined after approximately 15 days, where late solubilizers are distinguished by the formation of a solubilization halo detectable from the third day, while non-solubilizers lack this manifestation [7]. A transparent solubilization zone around bacterial colonies is a visual indicator of phosphate solubilization activity [8]. The solubilization index is defined as the ratio of the colony diameter to solubilization halo, termed phosphate solubilizing efficiency, and in solid culture media, it is expressed as the sum of the colony diameter (mm) plus the solubilization halo diameter (mm) divided by the colony growth diameter (mm) [9].

Traditional methods for the quantitative assessment of phosphate solubilization capacity, such as microscopy and plate counting [10,11], are laborious, time-consuming, and can be subjective, generating eye fatigue and inter-observer variability. Other methods, such as the Most Probable Number (MPN) [12], although useful for difficult-to-isolate bacteria, also have limitations in efficiency. Due to these limitations, there is a need to implement more efficient, accurate, and objective methods for the determination of phosphorus solubilization capacity [13,14].

In this sense, adapting different methods for determining solubilization capacity has allowed machine vision to play an important role in this area [15]. Specifically, bacterial counting using machine vision identifies patterns, dimensions, and key features that offer a level of accuracy and consistency that surpasses conventional methods [16].

For example, the research of Peña Cortés et al. in [17] proposes a machine vision system that uses high-resolution images to detect colonies in Petri dishes. Image segmentation is performed using color transformation and thresholding techniques. An approximate background map is created by erosion and subtraction to facilitate colony identification, obtaining a binary image in which colonies are distinguishable, thus simplifying the counting and characterization process. Likewise, in [18], bacteria were classified using multi-angle light scattering measurements from flow cytometry and the automated detection of bacterial communities using the BARDOT technique to classify bacterial species and strains. The results of the classifiers were cross-validated. Although the accuracy of bacterial species classification was high, the scope of this approach was limited, successfully identifying only a small number of species.

Ma et al. [19] developed a rapid method to detect Escherichia coli in food using artificial intelligence and optical imaging. The study employed YOLOv4 to identify microcolonies of Escherichia coli. YOLOv4 was trained to differentiate Escherichia coli from seven other common foodborne bacteria with an average accuracy of 94%. The system also allowed the rapid quantification of Escherichia coli concentrations and demonstrated a low false negative rate.

Despite studies applying machine vision for the identification or classification of various microorganisms, a knowledge gap persists in optimizing the process of determining the solubilization index of these bacteria using machine vision. Therefore, the main objective of this study is to optimize the determination of the solubilization index of phosphate-solubilizing bacteria by implementing a machine vision system, addressing the limitations, particularly the subjectivity and laboriousness, of traditional visual assessment methods. Therefore, unlike other vision systems focused on colony counting or bacterial identification, IGLOO (Image seGmentation anaLysis Of sOlubilization) offers a unique advantage with its dual segmentation capability for both the bacterial colony and its solubilization halo. This feature enables the direct and automated calculation of the solubilization index, a crucial metric in phosphorus biogeochemistry, thereby providing empirical evidence to support the study domain.

## 2. Materials and Methods

This section of the paper describes the development of IGLOO, a system for estimating the relative solubilization index of phosphate-solubilizing bacteria using machine vision techniques and neural network modeling (Figure 1). This system is primarily referenced by Rajwa et al. in [18], which implements a vision system using machine learning and pattern recognition for the unsupervised classification of bioparticles. This constitutes a structured pipeline that combines microbiological protocols and advanced machine vision techniques.

For the investigation, two phosphate-solubilizing bacterial strains, Enterobacter R11 and FCRK4, were grown on an NBRIP agar medium to induce colony growth and halo formation. To standardize the acquisition of images of bacterial colonies, a controlled imaging environment was constructed, including a light-isolated chamber with uniform LED illumination. Subsequently, the images were labeled using Roboflow [20] to generate a labeled dataset, which was used to train a YOLOv8 semantic segmentation model [21] configured with 200–500 epochs, a batch size of 4, and an input resolution of 800 × 800 pixels. Model training and optimization were performed in Google Colab using GPU acceleration. Finally, validation metrics were obtained from confusion matrices, while the effectiveness of the system in determining solubilization efficiency was compared with manual measurements of colony diameters and halos, ensuring an assessment of both technical performance and practical applicability.

### 2.1. Cultivation of R11 and FCRK4 Bacteria

The study population consisted of endophytic phosphate-solubilizing bacteria, previously characterized for their potential as biofertilizers, given that most of the phosphorus reserves in the soil are not in forms that plants can assimilate [22]. It worked with two bacterial strains, R11 and FCRK4, known as Plant Growth-Promoting Microorganisms (PGPB) [23].

The study population was extracted from Hacienda Los Naranjos, located in Cajibío (Cauca, Colombia), native to the bourbon coffee variety grown in andisol soils of the Popayán plateau. The bourbon variety coffee root samples, located at an altitude of 1870 m above sea level, with coordinates 2°35′11.6″ N, 76°33′11.2″ W [22], located near the Puracé volcano on volcanic soils, at a distance of 28 km from the city of Popayán, has characteristics of flat topography [24], with an ambient temperature of 12 to 18 °C [25].

#### Inoculation of Bacteria R11 and FCRK4

For the culture and inoculation process, bacterial strains extracted from Hacienda Los Naranjos and stored in the biology laboratory of Corporación Universitaria Comfacauca-Unicomfacauca under cryopreservation in 20% glycerol at −20 °C were reactivated in liquid culture medium M66; the preparation of the culture medium consisted of dissolving 15 g of the medium in 1 L of distilled water. Subsequently, 500 mL of the culture medium was poured into two Erlenmeyer flasks, which were partially closed with aluminum foil and cotton before autoclaving. Once sterilized, the M66 medium was allowed to cool to room temperature. The medium was then inoculated with an aliquot of the cryopreserved culture. The inoculated cultures were incubated at 28 °C for 24 h under continuous shaking conditions, consistent with the methodology employed in previous studies [26].

Once the strains were grown on the M66 culture medium, they were inoculated onto the NBRIP culture medium, developed to visualize phosphate solubilization [27] to estimate the solubilization capacity. The preparation of the NBRIP culture medium included the following components: 15 g yeast extract, 10 g glucose, 5 g Ca_3_(PO_4_)_2_, 5 g MgCl_2_ 6H_2_O, 0.25 g MgSO_4_ 7H_2_O, 0.2 g KCL, 0.1 g (NH_4_)_2_SO_4_ [28]. Likewise, the culture media were sealed with aluminum foil and autoclaved to avoid contamination or false positives. Finally, 25 mL of the culture medium was distributed in each Petri dish, and an aliquot of the culture medium was seeded in the dishes and incubated at 28 °C to verify the phosphorus solubilization capacity.

### 2.2. IGLOO Machine Vision System Software Development

#### 2.2.1. Dataset Construction

A fundamental aspect of developing image segmentation models is the availability of a suitable dataset for training. In this study, the dataset for model training was generated ad hoc (Figure 2) from images of bacterial inoculation in a controlled illumination environment due to the paucity of pre-existing datasets on this subject. The images were preprocessed and standardized to 1746 × 1746 pixels, with a resolution of 72 pixels per inch horizontally and vertically, under controlled illumination conditions. The imaging system utilized a 12-megapixel camera featuring an aperture of ƒ/2.8, a focal length of 77 mm, and a six-element lens. Optical image stabilization was achieved through sensor shift technology. The camera was consistently positioned 30 cm away from the Petri dish for all captured images.

Once standardized, the images were split using the “test and train” training approach [29], separating the dataset into a percentage to train the model and another to validate its efficiency. In this case, 75% of the images were used for training and 25% for testing, as proposed in other studies [30] and by the Common Objects in Context-COCO model guidelines for datasets. Although COCO has fewer categories than other datasets, it has more instances per category, facilitating learning detailed object models with accurate 2D localization [31,32].

Information labeling must be learned with YOLO algorithms to prepare the image dataset. For this, the precise delimitation of the objects employing adjusted bounding boxes (Figure 3) is crucial to guarantee the quality and efficiency of the training. The Roboflow platform [33] was employed for labeling, using an inter-observer agreement between two people to ensure agreement and quality in this process.

#### 2.2.2. IGLOO Training

For the training of the IGLOO system, convolutional neural networks (CNNs), architecture designed to process data with matrix structure [34], were used, which are key features for detecting and segmenting objects accurately required for the research objective. The training was performed using the open-source library YOLO version 8, which provides functions for segmentation, pose estimation, tracking, and classification [35]. Figure 4 shows the flowchart of the machine vision software developed in Google Colab, taking advantage of its cloud computing resources, such as the GPU, eliminating the need for expensive hardware [36].

The YOLOv8 framework from the Ultralytics library was used to train the IGLOO model, which offers a more advanced deep learning-based architecture than other models. First, the Ultralytics package was installed via pip, and the necessary libraries for image manipulation and visualization were imported. The YOLOv8 model was instantiated by specifying a YAML configuration file (yolov8n-seg.yaml) or using pre-trained weights (.pt). The training and validation data, organized into two classes (hereafter, ‘BAC’ refers to bacterial colonies and ‘AS’ to the detected solubilization halos), were managed using Roboflow [37]. This platform simplifies the segmentation and structured storage of datasets and provides access to training and validation data, allowing the direct import of paths into segmented image sets.

The training phase (Figure 5) was set up with a variable number of epochs to find the optimal value, a batch size of 4, and an image resolution of 800 × 800 pixels, which ensured that the morphological details of the bacterial colonies were effectively captured. The number of epochs was selected to allow adequate model convergence and the deep learning of relevant features, balancing the minimization of error and risk of overfitting [38]. The batch size of 4 images per iteration was chosen considering computational memory constraints in an environment such as Google Colab and the need for efficient gradient updates. A small batch size allows for more efficient memory use, although it may introduce more variability in model weight updates. On the other hand, the 800 × 800-pixel image size was selected to ensure sufficient resolution for the accurate detection and segmentation of bacterial colonies and solubilization areas without excessively increasing computational load and training times. Finally, the dataset was partitioned into training (75%) and test (25%) subsets to minimize overfitting and enhance the model’s generalizability. This strategy enabled the assessment of model performance on unseen data during training. The YOLOv8 model was initialized with pre-trained weights from the COCO large-scale dataset, employing transfer learning as an effective regularization technique. This approach leverages pre-learned features, reducing the likelihood of overfitting the task-specific dataset. The standard training procedure included the default weight decay value (L2 regularization) provided by Ultralytics. In addition, model performance on the validation set was monitored over several training epochs. This facilitated the selection of models that exhibited good generalization, thereby implicitly mitigating significant overfitting.

After training, the model was loaded using the saved weights to make predictions on new images. These predictions employed a 30% confidence threshold, balancing sensitivity and accuracy. The generated segmentation masks were converted into NumPy matrices for easy manipulation, allowing the area of the “BAC” and “AS” regions to be calculated in pixels. This quantitative analysis was important for determining the solubilization index, the central objective of the IGLOO study.

## 3. Results

The artificial vision system’s validation process for analyzing phosphate-solubilizing bacteria is divided into two stages. The first stage validates the detection of the bacteria and their solubilization halo. In this stage, the model receives new images, and its accuracy in detecting the objects of interest is evaluated (Table 1). The second stage compares the results obtained by the model with the traditional manual visual measurement technique used for estimating the solubilization index on agar plates.

### 3.1. Evaluation of IGLOO Model

For the evaluation of the model, multiple trainings were performed by varying the number of epochs to determine the optimal amount needed to predict the desired parameters. Seventy-five percent of the images were used for training and twenty-five percent for validation, following methodologies from previous research [39]. The results were evaluated experimentally by comparing a single image across a different number of epochs. The percentage of prediction accuracy and the confusion matrix corresponding to each training were analyzed, facilitating a clearer understanding of the model performance [40].

The results obtained with 10 epochs showed low accuracy. Figure 6 shows a test image and the predictions made by IGLOO, where it is observed that the solubilization halo, a crucial element to determine the solubilization rate, is not detected. As for the identification of bacterial colonies, the model identifies them partially but with low precision, considering the presence of three bacteria. The model trained with 10 epochs presented numerous errors so that nine true positives, eight false positives, seven false negatives, and thirty-five true negatives were identified for the class ‘AS’. The model’s accuracy for this class was 0.52, with a recall of 0.56 and an F1 score of approximately 0.54, indicating an imbalance between accuracy and recall.

With 50 epochs (Figure 7), the presence of the solubilization halo was observed with 55% confidence, although still with low certainty. The model identified the colony-forming units as a single element with 76% confidence, correctly assigning them to the ‘BAC’ class. Accuracy for the ‘AS’ class was 0.85, with a recall of 0.87 and an F1 score of 0.78, suggesting reasonable performance but with room for improvement. For the ‘BAC’ class, the accuracy was 0.85, recall 0.82, and F1 score 0.83, indicating a consistent and balanced performance.

The model demonstrates consistent performance with 100 training epochs (Figure 8), although some prediction errors persist. Compared to previous results, double prediction is observed in the “AS” class, which incorrectly includes part of the background. In addition, the confidence in predicting the “BAC” class decreased considerably. Unlike the previous 100 epochs, the model sometimes identifies multiple items for the same class during these runs. In the confusion matrix corresponding to this training, the “AS” class has 15 true positives, 3 false positives, 1 false negative, and 31 true negatives. For the “BAC” class, 23 true positives, 3 false positives, 5 false negatives, and 19 true negatives are recorded. Despite these problems, the model has shown progress, reaching an accuracy of 92% for “AS” and 84% for “BAC”, with F1 scores equal to or higher than 85% for both classes, confirming its improvement in learning.

At 200 epochs (Figure 9), the model shows higher confidence in classifying and predicting the desired elements in the images. Compared with the results of the model trained with 100 epochs, a better segmentation of the ‘BAC’ class is observed with 86% confidence in prediction. These results confirm the hypothesis that more epochs improve the model’s confidence. For these 200 epochs, the model identifies only one element of each class. In the confusion matrix associated with this training, there are 15 true positives, 3 false positives, 1 false negative, and 29 true negatives for the class ‘AS’. The ‘BAC’ class has 25 true positives, 1 false positive, 3 false negatives, and 19 true negatives.

The last trained model was the 500-epoch model, which presented the best confidence rate in predicting the ‘BAC’ and ‘AS’ classes. It is observed that a higher number of epochs significantly improves the results, although the risk of overfitting must be considered. In the case of this study, the 200- and 500-epoch models were used to contrast the results obtained when calculating the solubilization index. Figure 10 presents the confusion matrix corresponding to the 500-epoch training. There are no false positives for the ‘BAC’ class, indicating an accuracy of 100%. This class also has 23 true positives, 5 false negatives, and 18 true negatives. The ‘AS’ class has 15 true positives, 2 false positives, 1 false negative, and 28 true negatives.

Comparing the 200-epoch training to the 500-epoch training, improvements in accuracy and recall are observed, although some limitations persist in the ‘AS’ class. Accuracy went from 90% to 94%, maintaining a constant F1 score of 0.87. Accuracy for the ‘AS’ class remained at 0.83, suggesting that false positives were not significantly reduced despite additional training. For the ‘BAC’ class, recall remained at similar levels of 0.89 at 200 epochs and 0.82 at 500, indicating that false negatives remained challenging. It is concluded that it is feasible to refine accuracy percentages through higher levels of training. However, other strategies, such as dataset balancing or decision threshold adjustments, must be employed to improve recall rates. Despite these limitations, the behavior of the models satisfactorily meets the objectives set for determining the solubilization index. Finally, it is important to note that the lack of a significant reduction in false positives between 200 and 500 training epochs is likely due to practical limits imposed by the inherent visual ambiguity of solubilization halos in the current dataset and the specific learning dynamics of the YOLOv8 architecture, so these limitations should be worked on in future versions of IGLOO.

The performance metrics presented in Table 1 were derived from a single training run for each epoch configuration, selected to represent the behavior observed during development. Due to the considerable computational burden of training the IGLOO model, performing multiple complete runs for each epoch evaluation point was infeasible for this study. Nevertheless, the trends observed in Table 1 are clear, where the F1 score for the AS class shows a significant increase from 0.75 to 0.87, while the accuracy for BAC consistently increases to 1.00. These differences and the stabilization observed in several metrics after the 100–200 epochs suggest that the reported effects reflect learning patterns and not random fluctuations.

### 3.2. Comparison for Validation Between Traditional and IGLOO Methods

The solubilization index is a crucial parameter for assessing the ability of various phosphate-solubilizing organisms [41]. This parameter is routinely determined by a qualitative analysis that measures the size of the halo zone around the colony on a culture plate. Many researchers continue to use this plate assay method to assess organisms’ phosphate (P) solubilization capacity by observing the formation of a clear halo around the colony. However, the reliability of this method may vary depending on the researcher [42]. Researchers typically measure colony diameters and total diameters (including halos) manually. The halo size is obtained by subtracting the colony diameter from the total diameter [43]. An approximation of manual measurements to determine the colony diameter and halo diameter is presented in Figure 11.

The manual measurement of the colony diameter resulted in a value of 1.6 cm, while the solubilization halo diameter was determined to be 2.8 cm. To ensure the reliability of these measurements, an expert validation process was implemented, which allowed a comparative basis to be established with the results obtained by the IGLOO model. The same image used for the manual measurement was used to train the IGLOO deep learning model for 200 and 500 epochs (Figure 12). As demonstrated in the previous section, this training showed the highest levels of predictive confidence, justifying its selection for assessing the solubilization index. The trained models yielded favorable results for the prediction of colony forming units (CFUs) and the solubilization halo (HS), with 97% and 93% confidence for the 200-epoch model (Figure 12A) and 92% and 97% for the 500-epoch model (Figure 12B). These results indicate good performance compared to the previous training, suggesting that binary masks provide more detailed information for determining solubilization efficiency.

The corresponding binary masks were extracted after image segmentation, and the regions of interest were predicted. From these masks, quantitative parameters such as the area of the segmented regions were determined using the developed software. In the context of this study, the colony diameter and solubilization halo, essential parameters for the calculation of solubilization index, were measured. The data derived from the binary masks were stored in matrix structures, allowing algebraic data analysis. Importantly, these data represent the number of pixels corresponding to the segmented areas. A conversion from pixels to centimeters was performed to express the measurements in length (cm) units, using the known diameter of the Petri dish as a reference. The analyzed image had a width of 1746 pixels, equivalent to a Petri dish diameter of 11 cm. This ratio established a conversion factor of 158 pixels per centimeter. This factor allows the measured dimensions to be expressed in centimeters to calculate the solubilization index in conventional units. However, since the solubilization index is a dimensionless index, the direct calculation of this index from the pixel measurements extracted from the binary masks was also considered (Figure 13).

For the 200-epoch model, a pixel size of 13,329 was obtained for the segmented colony and 25,571 for the solubilization halo. The sizes for the 500-epoch model were 13,331 for the colony and 25,409 for the solubilization halo. In the final stage of IGLOO validation, the solubilization efficiency is calculated by comparing these data with the manually obtained efficiency. Equation (1) is used to determine this metric.Ef = (A + B)/A,(1)

A is the colony diameter, and B is the diameter of the solubilization halo [44]. The value of Ef obtained by the traditional manual visual measurement was 2.75. Both models (200 and 500 epochs) agreed highly with the manual reference value for automated segmentation. The 200-epoch model yielded a solubilization index of 2.92. This value remained identical when using the areas segmented directly into pixels and applying a pixel-to-centimeter conversion. Similarly, the 500-epoch model also resulted in an Ef of 2.91, regardless of direct pixels or conversion to centimeters.

The structured framework provided by IGLOO achieved a sub-6% difference relative to manual quantification, demonstrating efficacy in estimating solubilization efficiency. The consistency between training durations (200 and 500 epochs) and traditional manual visual measurement suggests high reproducibility, positioning IGLOO as a viable alternative to complex manual measurements.

## 4. Discussion of Results

The development of IGLOO represents a significant advance by automating the analysis of phosphate-solubilizing bacteria on solid media, addressing critical limitations of traditional manual visual evaluation techniques (halo zone measurements on plates). These manual methods are often characterized by subjectivity, lower throughput, and higher inter-observer variability [45]. This research introduces a machine vision system that leverages artificial intelligence to overcome conventional methods’ subjectivity and laborious nature for determining phosphate solubilization efficiency. With a difference of less than 6% over manual measurements, IGLOO’s performance exceeds that of these traditional techniques, providing a scalable solution for high-throughput screening, a need widely recognized in the literature [46]. Human errors can significantly affect measurement accuracy. Research indicates that these errors introduce variability in solubilization efficiencies, complicating the comparison of experimental results [42].

Implementing YOLOv8 in IGLOO, a state-of-the-art deep learning algorithm, represents a significant technological advance in this domain. IGLOO significantly accelerates the analysis process. Manual measurements can take several minutes per plate and hours for large datasets, whereas IGLOO processes images more rapidly, facilitating high-throughput estimation. This efficiency is crucial for large-scale studies, such as screening bacterial strains for biofertilizer development, where time savings can expedite research timelines [47]. The model was trained over multiple epochs (10, 50, 50, 200, and 500), evidencing a systematic improvement in detecting and segmenting bacterial colonies and solubilization halos. Thus, the progressive improvement in predictive confidence, particularly evident in the 200- and 500-epoch models, highlights the potential of artificial intelligence in microbiological image analysis, which aligns with previous research findings [48,49].

In this sense, IGLOO could have promising applications in the agricultural field for screening efficient phosphorus-solubilizing bacterial strains as biofertilizers [50], in environmental science for the study of nutrient cycling [47], and in biotechnology for quality control [51]. Specifically, IGLOO can be integrated into existing laboratory workflows for biofertilizer development, automating the manual process of reading and analyzing plates. This automation addresses a significant bottleneck in agricultural research. Additionally, IGLOO is valuable for quality control in biotechnology, particularly in biofertilizer production. In environmental sciences, IGLOO enhances soil health monitoring. Traditional methods for assessing soil phosphorus rely on chemical analysis. IGLOO, however, can be incorporated into soil sample analysis processes, offering a rapid, image-based assessment of phosphate solubilization potential.

However, despite its encouraging results, the evaluation of IGLOO was restricted to a limited set of strains with specific characteristics, suggesting the need for further validation. Also, factors such as image quality and the possible need for model re-training to adapt to different experimental conditions should be considered. Future research could focus on expanding the dataset used by implementing active learning strategies to optimize sample labeling [52]. In addition, exploring alternative or complementary deep learning architectures to YOLOv8 could improve performance. For example, U-Net and its variants, recognized for their success in biomedical image segmentation [53], could provide a more accurate delineation of colony boundaries and halos. An investigation into transformer-based segmentation models, such as SegFormer [54], or exploration of ensemble methods that integrate predictions from different models could also increase the robustness of IGLOO. In addition, expanding the dataset used and exploring the integration of IGLOO with laboratory instruments to provide real-time monitoring could enhance its robustness and applicability.

## 5. Validity Threats

While the IGLOO system demonstrates promising results for the automated determination of the solubilization index in phosphate-solubilizing bacteria, it is important to recognize potential threats to the validity of the study results.

Generalization to varying conditions: The system was validated using images acquired under uniform LED illumination and standardized resolutions. Thus, its performance may degrade in environments with varying illumination, lower-resolution cameras, or alternative imaging configurations.Scalability to field applications: This study focused on in vitro cultures. Implementing IGLOO in in situ agricultural environments, where soil particles, debris, or mixed microbial communities may clog colonies, presents unresolved challenges.Dataset limitation: The model was trained on a dataset derived from two bacterial strains (Enterobacter R11 and FCRK4) grown under controlled laboratory conditions. This limited diversity may restrict the model’s ability to generalize to other bacterial species or strains with distinct morphologies or solubilization patterns. To mitigate this, it is recognized that the dataset should be expanded to include diverse bacterial species, growth conditions, and imaging configurations.Overfitting Risk: Although more epochs improved performance, the marginal gains at 500 epochs and the persistence of false positives/negatives suggest a risk of overfitting. However, cross-validation was implicit in the training–test split.Variability of manual measurements: Manual measurements used for validation are based on expert judgment, which may introduce inter-observer variability. To mitigate this, it is important to validate IGLOO against multiple methods, such as spectrophotometry, in future studies to account for possible biases in manual measurements.Limited Scope of Comparative Validation Methods: The primary validation of IGLOO’s solubilization efficiency estimation was performed by comparing its results with those of the traditional manual ruler-based method of measuring colony diameters and halos on agar plates. Although IGLOO demonstrated significantly improved consistency and reduced subjectivity compared to this visual assessment technique, this study did not include direct comparisons with other established quantitative methods, such as colorimetric assays or inductively coupled plasma analysis that measures the actual concentration of solubilized phosphate in liquid or agar extracts. Therefore, although IGLOO effectively automates and standardizes the visual plate assay, its quantitative accuracy relative to the direct chemical measurement of solubilized phosphate requires further investigation in subsequent studies.

## 6. Conclusions

The present study describes the development and validation of the IGLOO machine vision system for the automated determination of the solubilization index of phosphate-solubilizing bacteria, addressing the limitations of traditional manual visual measurement. The results showed that IGLOO, trained with the YOLOv8 model under 200- and 500-epoch configurations, achieved greater than 90% accuracy in bacterial colony segmentation (“BAC” class) and solubilization halos (“AS” class), with a relative error of less than 6% compared to manual measurements. Supported by metrics such as an F1 score of 0.87–0.90 and a concordance in solubilization efficiency (2.91–2.92 vs. 2.75 manual), it evidences the system’s ability to reduce the subjectivity and variability associated with conventional techniques. The fundamental value of the proposed system lies in providing an objective, efficient, and reproducible quantification method, minimizing inter-observer variability.

In this sense, implementing IGLOO optimizes time and resources by eliminating reliance on visual analysis and improving scalability for large-scale studies in agricultural soils. The accurate conversion of pixel data to physical units using a calibrated factor allows the system to be applied in real environments, which, together with PGPB strains for biofertilizers, can play an important role in improving the domain.

While the model showed a slight risk of overfitting in prolonged training (500 epochs), the consistency among validation metrics suggests it is robust. In this regard, IGLOO offers significant advantages over conventional methods. Its ability to support the automation of the image analysis process reduces the time and effort required for phosphate solubilization assessment. In addition, the inherent objectivity of machine vision minimizes inter-observer variability, allowing for more consistent and reproducible results.

Future research could improve performance by balancing the dataset, including more bacterial species, and integrating data augmentation techniques to generalize the model. Finally, IGLOO represents a significant advance in the methodology for evaluating phosphate-solubilizing microorganisms. Its accuracy, efficiency, and objectivity make it a valuable tool for research in soil microbiology, plant nutrition, and biofertilizer development, allowing progress in sustainable agriculture.

## Figures and Tables

**Figure 1 microorganisms-13-00860-f001:**
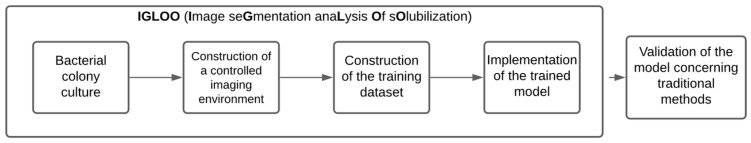
A methodological flowchart for the development and validation of the IGLOO system.

**Figure 2 microorganisms-13-00860-f002:**
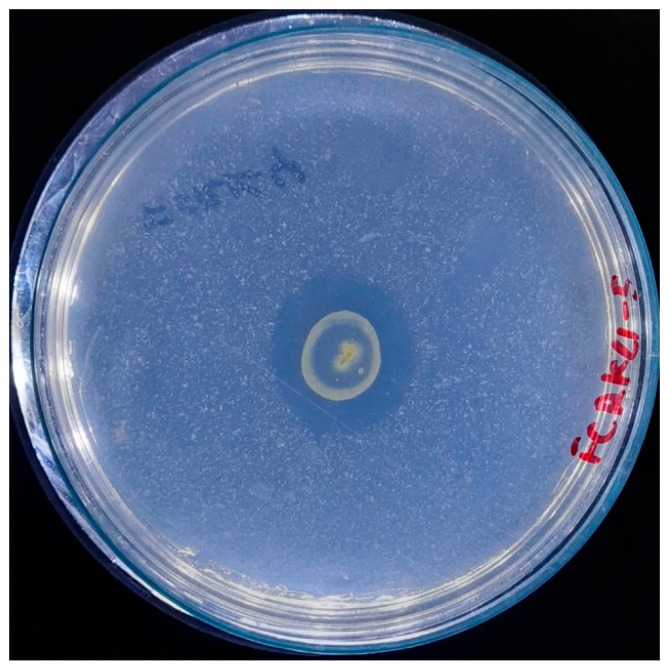
A representative standardized image was used in the IGLOO dataset, showing bacterial colonies.

**Figure 3 microorganisms-13-00860-f003:**
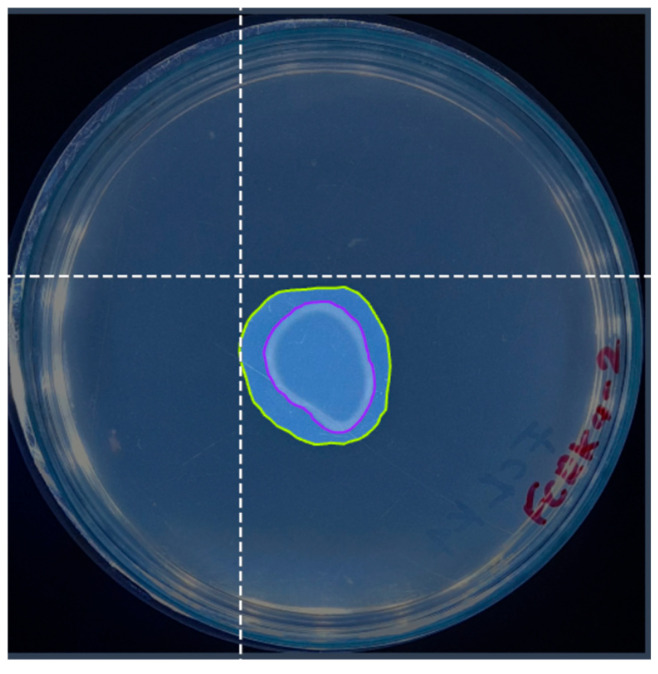
An example of image annotation using the Roboflow platform for semantic segmentation. The overlapping color masks precisely delimit the regions of interest, while the dotted line allows the annotation of relevant points.

**Figure 4 microorganisms-13-00860-f004:**
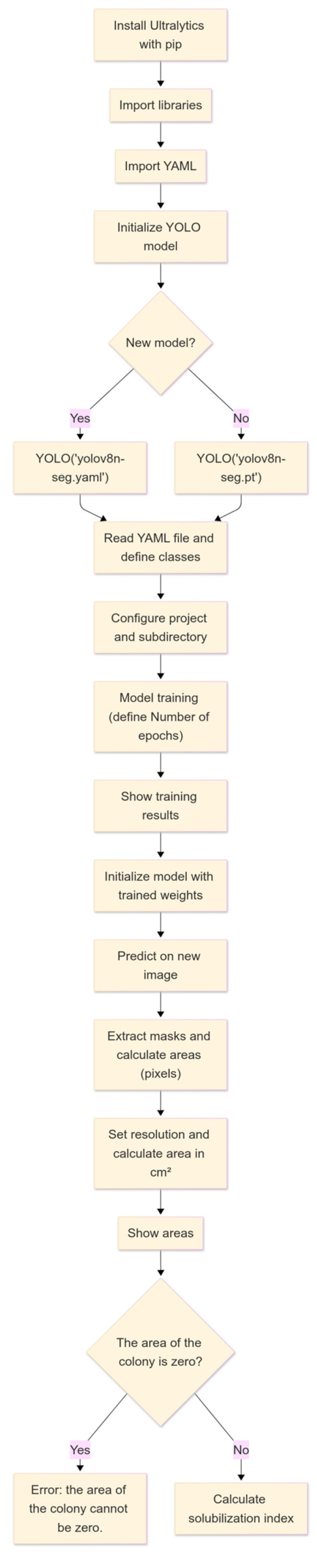
Operational flowchart of IGLOO software pipeline for automated solubilization index calculation.

**Figure 5 microorganisms-13-00860-f005:**
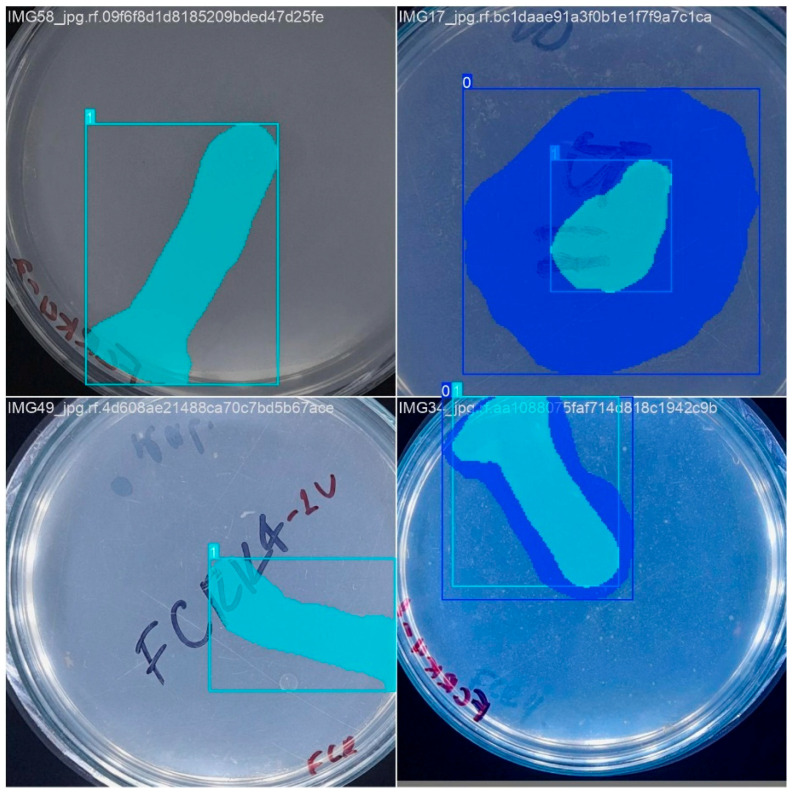
Representative examples of the training dataset illustrating the segmentation performed. The superimposed color masks delimit the bacterial colony and the corresponding solubilization halo.

**Figure 6 microorganisms-13-00860-f006:**
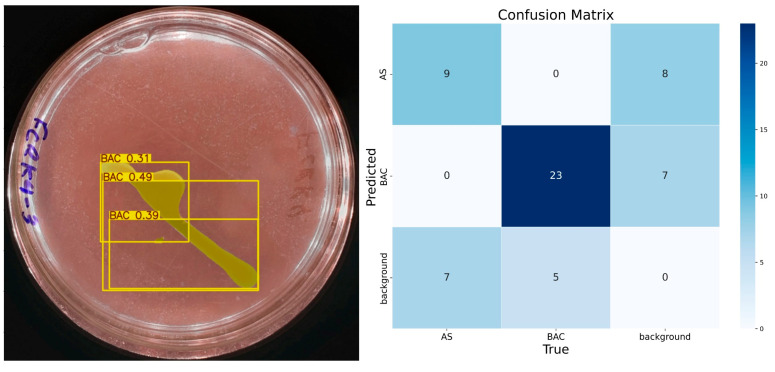
The performance of the IGLOO model after training for 10 epochs. (**Left**) Prediction on a sample test image. (**Right**) The confusion matrix for the test set.

**Figure 7 microorganisms-13-00860-f007:**
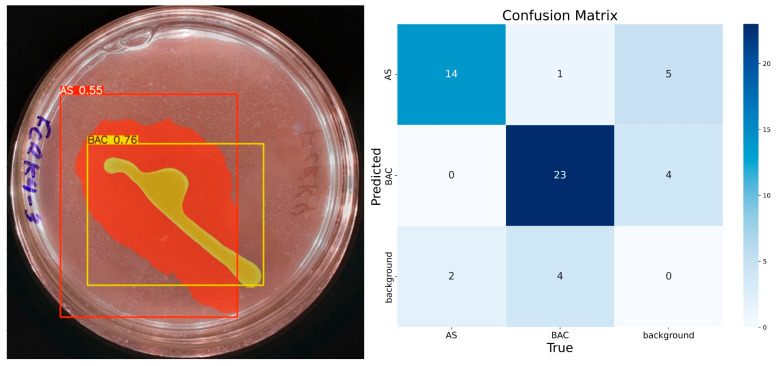
IGLOO model performance after training for 50 epochs. (**Left**) Prediction on the sample test image. (**Right**) The confusion matrix for the test set.

**Figure 8 microorganisms-13-00860-f008:**
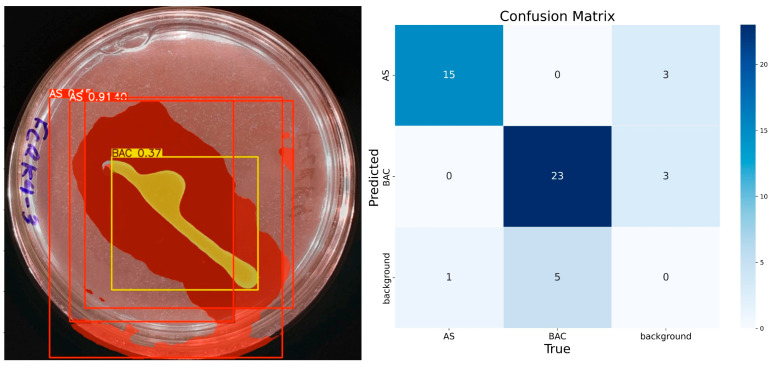
IGLOO model performance after 100 epoch training. (**Left**) Prediction on the sample test image. (**Right**) The confusion matrix for the 100-epoch test set.

**Figure 9 microorganisms-13-00860-f009:**
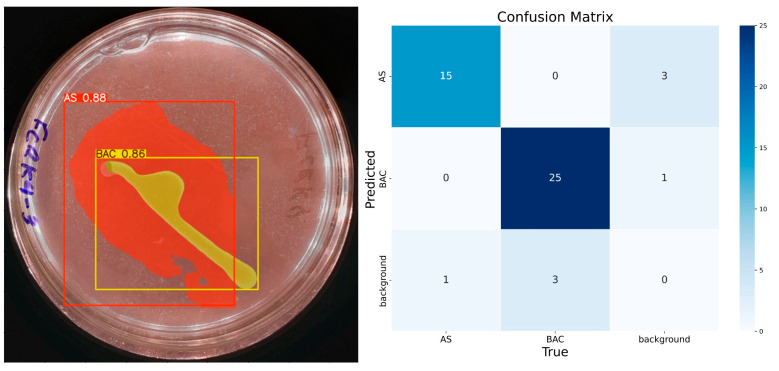
IGLOO model performance after training for 200 epochs. (**Left**) Prediction on the sample test image, showing the confident and well-defined segmentation of both the bacterial colony and the solubilization halo. (**Right**) The confusion matrix for the test set shows high accuracy.

**Figure 10 microorganisms-13-00860-f010:**
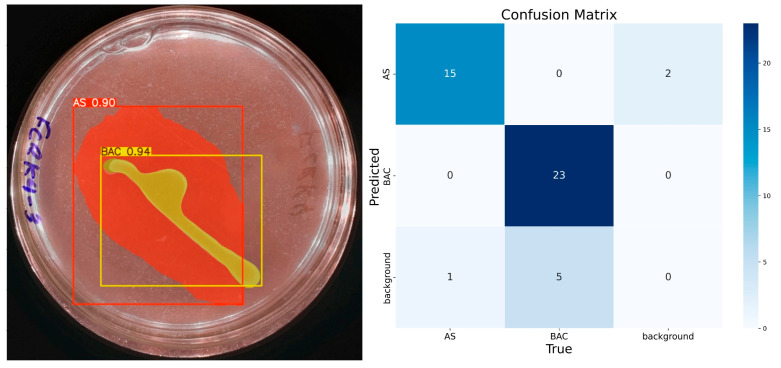
IGLOO model performance after 500-epoch training. (**Left**) Prediction on the sample test image, with high confidence segmentation for the colony. (**Right**) The confusion matrix for the test set confirms the model’s sustained high performance.

**Figure 11 microorganisms-13-00860-f011:**
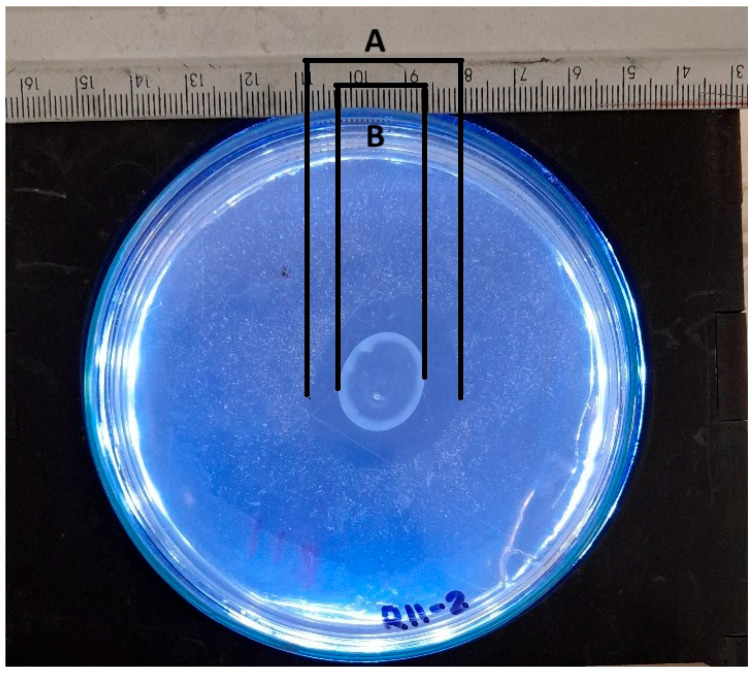
Schematic representation of manual measurement for determining solubilization index, where A represents the total diameter measurement, and B represents the colony diameter measurement.

**Figure 12 microorganisms-13-00860-f012:**
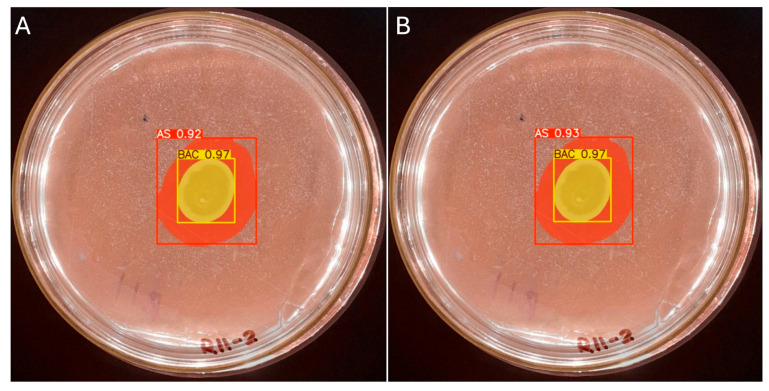
(**A**) Image segmentation for 200 epochs. (**B**) Image segmentation for 500 epochs.

**Figure 13 microorganisms-13-00860-f013:**
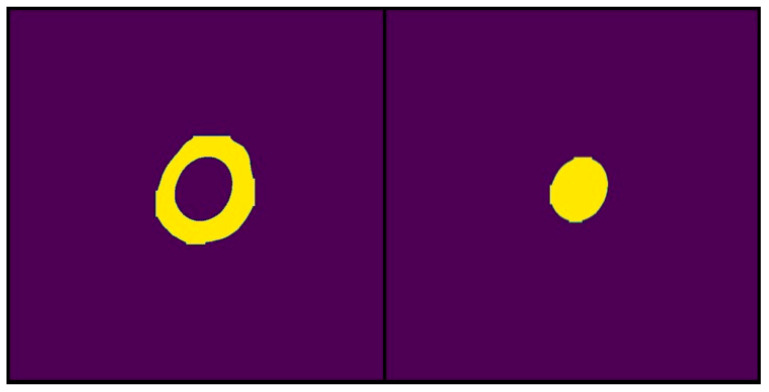
Examples of binary masks generated by IGLOO from the segmentation results used for area calculation. The yellow color highlights the pixels identified by the model as belonging to the segmented area. (**Left**) The binary mask represents the total area occupied by the solubilization halo and the colony. (**Right**) The binary mask represents only the area of the bacterial colony.

**Table 1 microorganisms-13-00860-t001:** Comparison of metrics of model trained at different epochs.

Metric	Class	10 Epochs	50 Epochs	100 Epochs	200 Epochs	500 Epochs
F1 Score	AS	0.75	0.78	0.87	0.87	0.87
F1 Score	BAC	0.72	0.83	0.85	0.9	0.9
Precision	AS	0.7	0.7	0.83	0.83	0.83
Precision	BAC	0.74	0.85	0.88	0.96	1
Accuracy	AS	0.78	0.85	0.92	0.91	0.93
Accuracy	BAC	0.76	0.83	0.84	0.91	0.89
Recall	AS	0.8	0.87	0.93	0.93	0.93

## Data Availability

Using the following link, you can access the material used to perform the bibliometric analysis and the literature review, as well as the materials used in this research (https://acortar.link/g8jRUv, accessed on 22 February 2025).

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
