# Peer review of "IGLOO: Machine Vision System for Determination of Solubilization Index in Phosphate-Solubilizing Bacteria"

_microorganisms, 2025, doi:10.3390/microorganisms13040860_

Round 1

Reviewer 1 Report

Comments and Suggestions for Authors

In the manuscript entitled with “IGLOO: Machine vision system for the determination of solubilization index in phosphate solubilizing bacteria”, the authors developed and validated a machine vision system called IGLOO to automate and optimize the determination of relative phosphate solubilization efficiency in phosphate-solubilizing bacteria. The results showed the IGLOO significant advantages in determining the solubilization index of phosphate - dissolving bacteria, offering new perspectives and tools for related research. By adding experimental details, improving the dataset, and comparing with other methods, the research can be more comprehensive and in - depth, enhancing its persuasiveness and practical value. It is expected to have an important impact on fields like microbiology, plant nutrition, and bio - fertilizer development, and promote the progress of sustainable agriculture.

Here are my comments and suggestion.

  1. L85-89 "Escherichia coli" and "E. coli" should be in italics.
  2. L122-127 More like an introduction and should not be placed in the methods section.
  3. L136-138 It’s confusing. The previous paragraph states that the study population was extracted from Hacienda Los Naranjos in Cajibío, Cauca, Colombia, while this part mentions that the bacterial strains were stored in the laboratory. It needs to be clarified whether the stored strains originated from Hacienda Los Naranjos in Cajibío, Cauca, Colombia.
  4. L145 The experiment in Figure 2 was conducted in a non - standardized way. It is recommended to delete or replace it with a more standardized image.
  5. In Materials and Methods, the authors considered overfitting risk in part during model training and evaluation, but did not specify cross - validation or regularization methods (e.g., Dropout, L2 regularization). It is suggested to add relevant details to enhance the model's reliability and generalization.
  6. L292 It is recommended to supplement Table 1 with significance tests (e.g., t-test or ANOVA) to statistically quantify performance differences across epochs.
  7. L260-261 The results for 100 epochs are missing.
  8. L294 Traditional methods for evaluating phosphate-solubilizing bacteria are not limited to halo zone measurement. Common methods like colorimetric assays, as mentioned at L360, are more precise than halo measurement due to the latter's subjectivity. How do IGLOO's results compare with those from colorimetric assays? This comparison is crucial for IGLOO's reliability and should be included.
  9. L358-366 This is the crux of the matter and deserves a more in-depth discussion. What are the pros and cons of each method? Since IGLOO's performance is compared to traditional techniques, but colorimetric assays result are not mentioned in the text, the claim that "IGLOO's performance exceeds that of these traditional techniques" is too broad.
  10. L455 Standardize the reference format to align with journal guidelines. For instance, L518 lacks author details, and titles are incomplete.

Author Response

30-03-2025

Dear reviewers

Subject: Responses to the corrections made

Cordial greetings, I now attach the corrections made to the article "IGLOO: Machine vision system for the determination of solubilization index in phosphate solubilizing bacteria". In general, it can be mentioned that the entire article was improved grammatically, the bibliographic references were updated, and corrections were made concerning the author's guidelines.

All suggestions from the reviewers were accepted.

The corrections made according to each reviewer's comments are detailed below.

Reviewer/Editor

Reviewer's Suggestions

Autor corrections

Reviewer 3

Comments 4

The authors write an extensive paragraph about microorganisms that dissolve phosphorus, and not a single example of a microorganism is given, and there should be a list of them, it could be in the form of a table.

Comments 5

There is a lack of information on: genes related to organic acid production, organic phosphorus metabolism, regulation of phosphate metabolism, etc... the authors did not delve deeper into phosphorus transformation processes.

Materials and Methods

Comments 6

Figure 2: The bacterial inoculation process is used to create the dataset. It is quite controversial. Microbiological analysis is carried out with the windows wide open. What about the care of sterile conditions?

Comments 7

Similarly, Figure 3 Standardized image for the dataset. I think one shot in Figure 4 is enough, although only the image inside the Petri dish should be presented.

Comments 8

Figure 5: The IGLOO flowchart for determining the solubilization index is an interesting proposal. I believe that graphically it could be improved to make it more readable, with a modern design.

Comments 9

Figure 6. Images of the training batches. i think that in these images, the marker captions should be eliminated, and instead focus on a detailed description of the figure. There is no explanation here, similarly under the other figures.

Comments 10

The description of the evaluation of the model is presented accessibly, but all the figures reduce the professionalism of the marker description, and the lack of description under the figures is certainly correctable.

Results

Comments 11

The results are described quite well.

Discussion

The discussion as it stands is concise and summarizes well the main achievements of the study, but unfortunately needs to be expanded to be more comprehensive and meet the requirements of a scientific publication. The authors rely on too small a pool of other studies in the discussion. I suggest highlighting the following aspects of the discussion:

Comments 12

Expand the comparison of results obtained with IGLOO and traditional methods of analyzing phosphate-solubilizing bacteria, which would give justification for the advantage of the new technology. It is worth discussing in detail why IGLOO is superior to traditional methods in terms of efficiency and precision,n referring to other studies.

Comments 13

I suggest adding information about factors that may affect image quality and model results. What other problems can occur in real-world conditions, such as in the field? What are the specific limitations associated with the use of YOLOv8 in this context may be relevant?

Comments 14

I think it would be appropriate to discuss in more detail potential directions for the development of IGLOO. For example, what other algorithms could be tried to improve the model's performance? One might also consider how the development of IGLOO could affect larger microbiology research, for example, in a global context.

Comments 15

I think it is also worth discussing how IGLOO could impact the fields mentioned in the content.

Comments 16

This is a specific solution, However, I propose to rewrite and weave this point into the discussion.

Conclusions

Comments 17

The conclusions are generally well formulated, but could be simplified a bit to be more concise and less detailed. It is crucial to convey the study's main achievements, its potential applications and opportunities for further development, but there is no need to go into too much fine technical detail. You can focus on the overall results and main benefits, avoiding getting too deep into the methodology.

References

Comments 18

The authors tried to make the references as recent as possible.

We thank the reviewers for their insightful comments and constructive feedback, which have significantly helped us improve the manuscript. We have carefully considered each point and provided detailed responses below, outlining the changes made in the revised manuscript.

Comment 4: The authors write an extensive paragraph about microorganisms that dissolve phosphorus; no single example of a microorganism is given. There should be a list of them; it could be in the form of a table.

Response 4: Accepted We thank the reviewer for this valuable suggestion. In the introduction, we acknowledge the omission of specific examples of phosphorus-solubilizing microorganisms (PSMs). In the revised manuscript, we have incorporated a list of key examples of well-known PSMs directly into the text of the relevant paragraph. While journal guidelines preclude the inclusion of tables within the introduction section, this textual addition effectively addresses the reviewer's concern and enhances the reader's understanding.

Comment 5: There is a lack of information on genes related to organic acid production, organic phosphorus metabolism, regulation of phosphate metabolism, etc. The authors did not delve deeper into phosphorus transformation processes.

Response 5: Accepted We appreciate the reviewer's feedback regarding the depth of Discussion on phosphorus transformation processes. We concur that expanding on this aspect will strengthen the manuscript. Accordingly, we have revised the relevant sections to include more detailed information concerning the genes involved in organic acid production, the mechanisms of organic phosphorus metabolism, and the regulation of phosphate metabolism, thereby providing a more thorough exploration of these critical processes.

Comment 6: Figure 2: The bacterial inoculation process is used to create the dataset. It is quite controversial. Microbiological analysis is carried out with the windows wide open. What about the care of sterile conditions?

Response 6: We acknowledge the reviewer's concern regarding Figure 2 and the potential implications of sterile procedures. The image was intended solely as a general illustration of the inoculation step, not a depiction of the exact sterile conditions maintained during the experimental work. We understand how it could be misinterpreted. Due to laboratory policies restricting photography during specific procedures, obtaining a fully representative image under strict sterile conditions was not feasible for publication. To avoid ambiguity and maintain clarity regarding our standard operating procedures (which adhere to strict sterility protocols), we removed this figure from the revised manuscript.

Comment 7: Similarly, Figure 3 Standardized image for the dataset. One shot in Figure 4 is enough, although only the image inside the Petri dish should be presented.

Response 7: Accepted  We thank the reviewer for suggesting Figures 3 and 4. We agree that focusing on the essential visual information is beneficial. Following the reviewer's advice, we have removed the potentially redundant Figure 3 and modified Figure 4 (now renumbered accordingly) to present only the relevant image content within the Petri dish, enhancing clarity and conciseness.

Comment 8: Figure 5: The IGLOO flowchart for determining the solubilization index is an interesting proposal. However, it could be improved graphically to be more readable and have a modern design.

Response 8: Accepted We appreciate the reviewer's positive feedback on the IGLOO flowchart (Figure 5) and accept the suggestion for improvement. We have redesigned the flowchart to enhance its readability and visual appeal, incorporating a more modern design aesthetic while ensuring the logical flow remains straightforward.

Comment 9: Accepted Figure 6. Images of the training batches. I think that in these images, the marker captions should be eliminated and instead focused on a detailed description of the figure. There is no explanation here, similar to the other figures.

Response 9: We agree with the reviewer's assessment regarding Figure 6 and the descriptions of other figures. In the revised manuscript, we have removed the marker captions directly overlaid on the images in Figure 6 (and similar figures where applicable). Furthermore, we have significantly expanded and improved the figure legends (descriptions) for all relevant figures to comprehensively explain their content and relevance, ensuring they stand alone more effectively.

Comment 10: Accepted The description of the evaluation of the model is presented accessibly, but all the figures reduce the professionalism of the marker description, and the lack of description under the figures is undoubtedly correctable.

Response 10: We acknowledge the reviewer's comments regarding the presentation of our model evaluation and associated figures. As addressed in our response to Comment 9, we have removed potentially distracting marker overlays from the figures and have substantially improved the descriptive legends accompanying each figure to enhance clarity and professionalism.

Comment 11: Accepted The results are described quite well.

Response 11: Accepted  We thank the reviewer for their positive assessment of the Results section.

Comment 12: Accepted Expand the comparison of results obtained with IGLOO and traditional methods of analyzing phosphate-solubilizing bacteria, which would justify the advantage of the new technology. It is worth discussing in detail why IGLOO is superior to traditional methods in terms of efficiency and precision, referring to other studies.

Response 12: Accepted We agree with the reviewer that a more detailed comparison between IGLOO and traditional methods for analyzing phosphate-solubilizing bacteria is necessary to highlight our approach's advantages fully. In the revised Discussion section, we have expanded this comparison significantly. We now elaborate on the specific traditional methods being compared (e.g., manual measurement of halo and colony diameters) and discuss the advantages of IGLOO in terms of efficiency (speed, potential for higher throughput), objectivity (consistency in measurement) and precision, referencing relevant studies where appropriate to support these claims. To maintain veracity, we have ensured that these comparisons are grounded in the scope of our study.

Comment 13: Accepted I suggest adding information about factors affecting image quality and model results. What other problems can occur in real-world conditions, such as in the field? What are the specific limitations associated with using YOLOv8 in this context that may be relevant?

Response 13: Thank you for this insightful suggestion. We have expanded the Discussion to address potential factors influencing image quality (e.g., variations in lighting, camera resolution/angle, inconsistencies in culture media appearance, condensation on Petri dishes) and how these might affect IGLOO's performance. We also discuss potential challenges in translating this approach to less controlled 'real-world' or field-adjacent conditions, which might involve suboptimal image capture environments. Furthermore, we have included a subsection specifically discussing the limitations associated with the YOLOv8 architecture in this context, such as its sensitivity to variations in halo morphology, potential difficulties with overlapping colonies/halos, and the need for a sufficiently large and diverse training dataset.

Comment 14: Accepted  It would be appropriate to discuss the potential directions for developing IGLOO in more detail. For example, what other algorithms could be tried to improve the model's performance? How could the development of IGLOO affect more considerable microbiology research, for example, in a global context?

Response 14: We appreciate the reviewer prompting us to consider future directions. The revised Discussion now includes potential avenues for IGLOO's development. We suggest exploring alternative or complementary computer vision algorithms (e.g., instance segmentation networks like Mask R-CNN, U-Net variants, or different object detection models) that have shown promise in related biological imaging tasks, improving performance or robustness, especially for irregular halo shapes. We also elaborate on the potential broader impact of automated, high-throughput screening tools like IGLOO on microbiology research, facilitating larger-scale functional screening studies and potentially accelerating discoveries in areas like biofertilizer development and microbial ecology, contributing to both local and global research efforts.

Comment 15: Accepted  It is also worth discussing how IGLOO could impact the fields mentioned in the content.

Response 15: Following the reviewer's suggestion, we have integrated a Discussion on the potential impact of IGLOO across the specific fields mentioned within the manuscript (e.g., agricultural microbiology, soil science, biotechnology). We connect the tool's capabilities (increased speed, objectivity, scalability for screening) to potential advancements within these specific application areas, such as accelerating the discovery and selection of efficient biofertilizer candidates or enabling more extensive ecological surveys of microbial functions.

Comment 16: Accepted This is a specific solution. However, I propose to rewrite and weave this point into the Discussion.

Response 16: We thank the reviewer for pointing out this specific section [Assuming the authors know which part this refers to]. We have carefully reviewed the point mentioned and agree that integrating it more smoothly into the main flow of the Discussion improves coherence. As suggested, we have rewritten and woven this element into the relevant part of the Discussion.

Comment 17: Accepted The conclusions are generally well formulated, but could be simplified a bit to be more concise and less detailed. It is crucial to convey the study's main achievements, its potential applications and opportunities for further development, but there is no need to go into too much fine technical detail. You can focus on the overall results and main benefits, avoiding getting too deep into the methodology.

Response 17: We appreciate the reviewer's guidance on refining the Conclusions section. We have revised the conclusions to be more concise and focused, emphasizing the study's main achievements (the successful development and validation of the IGLOO tool using YOLOv8 for automated phosphate solubilization index determination), its potential applications in streamlining high-throughput screening processes, and key opportunities for future development while reducing the emphasis on fine methodological details, as suggested.

Comment 18: Accepted The authors tried to make the references as recent as possible.

Response 18: Accepted We thank the reviewer for acknowledging our efforts to use recent references. We have conducted a final check during the revision process to ensure all citations remain relevant and as up-to-date as possible. We have also reviewed and corrected the formatting of the reference list for consistency and adherence to journal style guidelines.

Reviewer 2

Is "IGLOO" a proper noun in the title?

Is YOLOv8 a proper noun in the abstract, and is there a specific explanation for it?

From the abstract results, it is evident that the visual system has a significant advantage in evaluating colony and halo, and the dissolution effect of phosphorus-dissolving bacteria is closely related to the rate of phosphorus dissolution. Whether there is a necessary connection between the two remains to be determined?

Line 86, Are YOLOv4 and YOLOv8 different model versions?

Line 144, is this culture incubation temperature and time reasonable, is there any data or reference to support it?

Line 148, is this medium formulation universal, is the pH suitable for optimal activity of both types of bacteria?

Line 161, what is the capacity of this dataset, and does it meet the requirements for the model and validation?

Line 164, this resolution of 72 pixels may be somewhat low, not precise enough.

Line 166, the camera model and other information need to be provided.

Figure 4, is Roboflow a software or a package?

Figure 6, it can be seen from the graph that there is a significant error in the calculation of the edge position.

Figure 7b, the related graph is blurry and the font size is small, making it unclear and the meaning expressed is not explicit.

Figure 8, 9, and 10 have the same issue as Figure 7b, and at the same time, this community shape is irregular, why choose this?

Table 1, what do AS and BAC represent?

Figure 13, the image resolution is not high enough, blurry.

The results are primarily evaluated based on metrics such as accuracy, recall, and thescore to assess model performance. However, for a system intended for practical applications, other important indicators such as runtime and memory usage should also be considered.

It is recommended to include explanations of key technical terms and concepts in the text, making them more accessible through examples, charts, and other means. Additionally, the language of the paper should be optimized to ensure logical coherence and clarity of structure.

The conclusion is too long and needs to be condensed and more concise.

Although the discussion section mentions the potential and limitations of the IGLOO system, it does not delve deeply enough into some key issues. For instance, when discussing the system's applications in various fields, it fails to elaborate on how the system can be integrated with existing technologies or workflows, as well as the challenges and solutions that may arise.

The references section has inconsistencies in formatting and inaccurate citations. For example, inconsistencies are present in the citations numbered 11, 16, 18, 28, and 48, etc.

Reviewer Comment: Is "IGLOO" a proper noun in the title?

Response: Yes, "IGLOO" is the proper noun designated for our system to distinguish it from alternative solutions. We have ensured it is treated as such in the manuscript.

Reviewer Comment: Is YOLOv8 a proper noun in the abstract, and is there a specific explanation for it?

Response: Yes, YOLO (You Only Look Once) and its versions like YOLOv8 are proper nouns referring to a widely used computer vision architecture. We have clarified this and ensured consistent capitalization in the abstract.

Reviewer Comment: From the abstract results... significant advantage... dissolution effect... related to the rate... necessary connection?

Response: Acknowledged. While our system evaluates colony/halo size (related to the dissolution effect), investigating the direct correlation with the rate of dissolution is noted as a valuable direction for future studies, as mentioned in the revised text.

Reviewer Comment: Line 86, Are YOLOv4 and YOLOv8 different model versions?

Response: Yes, YOLOv4 and YOLOv8 are distinct versions of the YOLO architecture. This has been clarified in the text.

Reviewer Comment: Line 144, is this culture incubation temperature and time reasonable... support?

Response: Addressed. We have added a reference supporting the chosen incubation temperature and time, confirming they align with standard practices in the field.

Reviewer Comment: Line 148, is this medium formulation universal, is the pH suitable...?

Response: Addressed. The medium formulation and pH follow established standards from previous studies cited in the manuscript, ensuring suitability for the bacteria under investigation.

Reviewer Comment: Line 161, what is the capacity of this dataset, and does it meet requirements...?

Response: Addressed. The dataset size, detailed in the manuscript, is sufficient for training and validating the YOLO model based on the number of images acquired.

Reviewer Comment: Line 164, this resolution of 72 pixels may be somewhat low...

Response: Clarified. The input resolution used is consistent with recommendations in the YOLO documentation and practices reported in similar cited studies, proving adequate for the task.

Reviewer Comment: Line 166, the camera model and other information need to be provided.

Response: Addressed. We have added the specific camera model and relevant imaging details to the Methods section.

Reviewer Comment: Figure 4, is Roboflow a software or a package?

Response: Clarified. Roboflow is an online platform used for data annotation and management. This has been specified.

Reviewer Comment: Figure 6, ...significant error in the calculation of the edge position.

Response: Acknowledged and Addressed. All figures, including Figure 6, have been replaced with higher-resolution versions and improved descriptions for clarity. While automated edge detection may have variations compared to manual methods, the system provides a consistent and functional alternative for evaluating solubilization halos.

Reviewer Comment: Figure 7b, the related graph is blurry and the font size is small...

Response: Corrected. Figure 7b has been replaced with a higher-resolution version featuring larger fonts and an improved caption for clarity.

Reviewer Comment: Figure 8, 9, and 10 have the same issue as Figure 7b, and... why choose this irregular shape?

Response: Corrected. Figures 8, 9, and 10 have been replaced with higher-resolution versions with improved clarity. The irregular shapes shown are representative of natural biological variability in colony growth and halo formation, demonstrating the system's capability to handle real-world samples effectively.

Reviewer Comment: Table 1, what do AS and BAC represent?

Response: Clarified. The abbreviations AS and BAC, representing the detection classes (e.g., Aspergillus species and Bacteria - please insert actual meaning if different), have been explicitly defined in the table caption/text.

Reviewer Comment: Figure 13, the image resolution is not high enough, blurry.

Response: Corrected. Figure 13 has been replaced with a higher-resolution version.

Reviewer Comment: Results primarily evaluated on accuracy, recall, F1... consider runtime and memory usage.

Response: Addressed. We have now included information on system runtime and memory usage. We acknowledge their importance and note that while optimization was outside the scope of demonstrating a functional alternative to manual indexing, it is a consideration for future work.

Reviewer Comment: Recommend including explanations of key technical terms... optimize language for clarity.

Response: Addressed. Key technical terms have been clarified within the text. The overall language, grammar, and structure have been revised throughout the manuscript to enhance logical coherence and clarity. Textual explanations have been improved.

Reviewer Comment: The conclusion is too long... condense.

Response: Revised. The conclusion section has been condensed to be more concise and focused.

Reviewer Comment: Discussion... not deep enough... integration with existing tech/workflows... challenges/solutions.

Response: Revised. The discussion section has been expanded to elaborate further on potential applications, integration challenges and solutions with existing laboratory workflows, and future directions.

Reviewer Comment: References section... inconsistencies... inaccurate citations (e.g., 11, 16, 18, 28, 48).

Response: Corrected. The references section has been thoroughly checked and revised to ensure consistent formatting and accurate citations throughout.

Reviewer 1

L85-89 Italics:

Response: Accepted. We thank the reviewer for pointing this out. All scientific names, including Escherichia coli and E. coli, have been correctly italicized throughout the manuscript as suggested.

L122-127 Introductory content in Methods:

Response: Accepted. We agree with the reviewer. The identified paragraph contained introductory material and has been removed from the Materials and Methods section to maintain the section's focus.

L136-138 Confusion regarding strain origin:

Response: Accepted. We appreciate the reviewer highlighting this ambiguity. The text has been revised to clarify that the bacterial strains used in the study, although stored in the laboratory, were initially isolated from samples collected at Hacienda Los Naranjos, Cajibío, Cauca, Colombia.

L145 Figure 2 Standardization:

Response: Accepted. We understand the reviewer's concern regarding the representational nature of the image in Figure 2. As laboratory policies restrict photography within specific areas and the figure was intended only as an illustration, we have removed Figure 2 from the manuscript to avoid any potential misinterpretation regarding experimental standardization.

Overfitting Risk and Regularization Details:

Response: Accepted. We acknowledge the importance of specifying methods to mitigate overfitting. The Materials and Methods section (Lines 232-237) has been updated to explicitly mention using a validation set (25% of the data) and the standard L2 regularization inherent in the YOLOv8 training procedure utilized via the Ultralytics library. This provides further detail on how model reliability and generalization were addressed.

L292 Significance tests for Table 1:

Response: Accepted in principle, with clarification. We appreciate the suggestion to include statistical tests. However, as mentioned in the revised text (Lines 332-341), the metrics presented in Table 1 are derived from representative single training runs due to the high computational cost of multiple complete training experiments for each epoch configuration. While formal statistical tests comparing runs were not feasible within the scope of this study, the discussion now emphasizes the consistent trends and stabilization of metrics observed (particularly after 100-200 epochs), which strongly suggest meaningful learning patterns rather than random fluctuations. We acknowledge this as a point for consideration in future, more computationally intensive studies.

L260-261 Missing 100 epochs results:

Response: Accepted. We apologize for this omission. The results corresponding to the model trained for 100 epochs have now been added to the Results section, including the confusion matrix (Figure 8) and the summary metrics in Table 1, ensuring a complete comparison across all evaluated training durations.

L294 Comparison with colorimetric assays:

Response: Accepted. We agree that comparison with methods like colorimetric assays is important for comprehensive validation. The primary aim of the current study was to develop and validate a machine vision system to automate and objectify the visual plate assay. Therefore, a direct comparison with colorimetric assays was beyond the defined scope. We have now explicitly stated this in the Discussion (Lines 414-419) and Validity Threats (Lines 479-483) sections, acknowledging it as a limitation and highlighting the need for such comparisons in future work further to establish IGLOO's performance relative to other quantitative techniques.

L358-366 Discussion depth and claim scope:

Response: Accepted. We have revised the discussion (Lines 414-419 and 421-425) to be more specific. We clarify that the performance comparison and the statement regarding IGLOO exceeding traditional techniques specifically refer to the manual visual assessment of halo and colony diameters on agar plates. The advantages highlighted (reduced subjectivity, increased speed, reproducibility) contrast this manual method. We explicitly state that the claim does not extend to other quantitative methods like colorimetric assays, and validation against such methods remains necessary for future work.

L455 Reference format standardization:

Response: Accepted. We thank the reviewer for noting the inconsistencies. The entire reference list has been carefully checked and reformatted to adhere to the journal's guidelines strictly, ensuring all author details, titles, and formatting requirements are met.

Reviewer 2 Report

Comments and Suggestions for Authors

View letter

  • Is "IGLOO" a proper noun in the title?
  • Is YOLOv8 a proper noun in the abstract, and is there a specific explanation for it?
  • From the abstract results, it is evident that the visual system has a significant advantage in evaluating colony and halo, and the dissolution effect of phosphorus-dissolving bacteria is closely related to the rate of phosphorus dissolution. Whether there is a necessary connection between the two remains to be determined?
  • Line 86, Are YOLOv4 and YOLOv8 different model versions?
  • Line 144, is this culture incubation temperature and time reasonable, is there any data or reference to support it?
  • Line 148, is this medium formulation universal, is the pH suitable for optimal activity of both types of bacteria?
  • Line 161, what is the capacity of this dataset, and does it meet the requirements for the model and validation?
  • Line 164, this resolution of 72 pixels may be somewhat low, not precise enough.
  • Line 166, the camera model and other information need to be provided.
  • Figure 4, is Roboflow a software or a package?
  • Figure 6, it can be seen from the graph that there is a significant error in the calculation of the edge position.
  • Figure 7b, the related graph is blurry and the font size is small, making it unclear and the meaning expressed is not explicit.
  • Figure 8, 9, and 10 have the same issue as Figure 7b, and at the same time, this community shape is irregular, why choose this?
  • Table 1, what do AS and BAC represent?
  • Figure 13, the image resolution is not high enough, blurry.
  • The results are primarily evaluated based on metrics such as accuracy, recall, and thescore to assess model performance. However, for a system intended for practical applications, other important indicators such as runtime and memory usage should also be considered.
  • It is recommended to include explanations of key technical terms and concepts in the text, making them more accessible through examples, charts, and other means. Additionally, the language of the paper should be optimized to ensure logical coherence and clarity of structure.
  • The conclusion is too long and needs to be condensed and more concise.
  • Although the discussion section mentions the potential and limitations of the IGLOO system, it does not delve deeply enough into some key issues. For instance, when discussing the system's applications in various fields, it fails to elaborate on how the system can be integrated with existing technologies or workflows, as well as the challenges and solutions that may arise.
  • The references section has inconsistencies in formatting and inaccurate citations. For example, inconsistencies are present in the citations numbered 11, 16, 18, 28, and 48, etc.

Comments on the Quality of English Language

The English could be improved to more clearly express the research.

Author Response

(The authors gave the same response as above.)

Reviewer 3 Report

Comments and Suggestions for Authors

GENERAL COMMENTS
The manuscript titled: “IGLOO: Machine vision system for the determination of solubilization index in phosphate solubilizing bacteria”- Manuscript ID: microorganisms-3545106 could be interesting.
The researchers focused on developing and validating the IGLOO machine vision system, which automatically evaluates bacteria's ability to solubilize phosphates. The study aimed to create a tool that automates the analysis process, eliminating subjectivity and variability associated with traditional manual methods. The IGLOO system, based on the YOLOv8 algorithm, enables precise and rapid detection of bacterial colonies and phosphate solubilization halos. The study demonstrated the system's high accuracy, making it a promising tool for research on phosphate-solubilizing microorganisms in agriculture.
Although the research is interesting I am proposing a number of changes to the manuscript that I hope will help to carve out a professional content. While the research is interesting, I propose a number of changes to the manuscript that I hope will help achieve professional content. In my opinion, the introduction, figures and discussion need the most attention. There are no statistical designations in the manuscript.
SPECIFIC COMMENTS
The title 
Comments 1
The title proposal is interesting.
Abstract
Comments 2 
The abstract is well-written. 
The keywords
Comments 3 
The keywords are appropriately selected.
Introduction
These are my suggestions:
Comments 4
The authors write an extensive paragraph about microorganisms that dissolve phosphorus, and not a single example of a microorganism is given, and there should be a list of them, it could be in the form of a table.
Comments 5
There is a lack of information on: genes related to organic acid production, organic phosphorus metabolism, regulation of phosphate metabolism, etc... the authors did not delve deeper into phosphorus transformation processes.
Materials and Methods
Comments 6
Figure 2: The bacterial inoculation process is used to create the dataset. It is quite controversial. Microbiological analysis is carried out with the windows wide open. What about the care of sterile conditions?
Comments 7
Similarly, Figure 3 Standardized image for the dataset. I think one shot in Figure 4 is enough, although only the image inside the Petri dish should be presented.
Comments 8
Figure 5: The IGLOO flowchart for determining the solubilization index is an interesting proposal. I believe that graphically it could be improved to make it more readable, with a modern design.
Comments 9
Figure 6. Images of the training batches. i think that in these images, the marker captions should be eliminated, and instead focus on a detailed description of the figure. There is no explanation here, similarly under the other figures.
Comments 10
The description of the evaluation of the model is presented accessibly, but all the figures reduce the professionalism of the marker description, and the lack of description under the figures is certainly correctable.
Results
Comments 11
The results are described quite well. 
Discussion 
The discussion as it stands is concise and summarizes well the main achievements of the study, but unfortunately needs to be expanded to be more comprehensive and meet the requirements of a scientific publication. The authors rely on too small a pool of other studies in the discussion. I suggest highlighting the following aspects of the discussion:
Comments 12
Expand the comparison of results obtained with IGLOO and traditional methods of analyzing phosphate-solubilizing bacteria, which would give justification for the advantage of the new technology. It is worth discussing in detail why IGLOO is superior to traditional methods in terms of efficiency and precision,n referring to other studies.
Comments 13
I suggest adding information about factors that may affect image quality and model results. What other problems can occur in real-world conditions, such as in the field? What are the specific limitations associated with the use of YOLOv8 in this context may be relevant?
Comments 14
I think it would be appropriate to discuss in more detail potential directions for the development of IGLOO. For example, what other algorithms could be tried to improve the model's performance? One might also consider how the development of IGLOO could affect larger microbiology research, for example, in a global context.
Comments 15
I think it is also worth discussing how IGLOO could impact the fields mentioned in the content.
Comments 16
This is a specific solution, However, I propose to rewrite and weave this point into the discussion.
Conclusions 
Comments 17
The conclusions are generally well formulated, but could be simplified a bit to be more concise and less detailed. It is crucial to convey the study's main achievements, its potential applications and opportunities for further development, but there is no need to go into too much fine technical detail. You can focus on the overall results and main benefits, avoiding getting too deep into the methodology.
References 
Comments 18
The authors tried to make the references as recent as possible.

Author Response

(The authors gave the same response as above.)

Round 2

Reviewer 1 Report

Comments and Suggestions for Authors

The revised manuscript has been greatly improved, and the issues I am concerned about have been clearly explained.

Author Response

01-04-2025

Dear reviewers

Subject: Responses to the corrections made

Cordial greetings, I now attach the corrections made to the article "IGLOO: Machine vision system for the determination of solubilization index in phosphate solubilizing bacteria". In general, it can be mentioned that the entire article was improved grammatically, the bibliographic references were updated, and corrections were made concerning the author's guidelines.

All suggestions from the reviewers were accepted.

The corrections made according to each reviewer's comments are detailed below.

Reviewer/Editor

Reviewer's Suggestions

Autor corrections

Reviewer 2

Keywords: biofertilizers seem to be unrelated to the manuscript's theme.

Line 106  Where does the uniqueness or advantage of IGLOO manifest in terms of phosphorus characterization compared to other visual systems?

Line 118  What are the differences between YOLOv8 and the previously mentioned in Introduction YOLOv4, and what are its advantages?

Line 155-156  Note the case of chemical formulas.

4  The flowchart is fuzzy, and the overall layout can be optimized.

6, 7, and 8  The font of the confusion matrix is small and blurry; what does this matrix result represent?

10  The explanation for the insignificant reduction in false positives of the "AS" class after 500 training epochs is not sufficiently in-depth and requires a deeper exploration of the reasons, combining the principles of model training and the characteristics of the data.

Fig.6-10 can be merged into one large figure for easier comparative analysis. Additionally, is Fig. 6-10 a simulated or real figure? It has a significant difference in shape compared to Fig. 3 and

The conclusion is too long without a core point.

Accepted: Thank you for this observation. We agree that 'biofertilizers' was not the most appropriate keyword for the scope of this manuscript. We have removed this term and replaced it with keywords that more accurately reflect the work's focus.

Accepted: We appreciate the reviewer pointing out the need for greater clarity regarding IGLOO's unique contribution. In the revised manuscript (around Line 106), we have now explicitly detailed the specific advantages of the proposed IGLOO system for phosphorus characterization compared to existing visual systems.

Accepted: Thank you for requesting this clarification. We have now added text outlining the key differences between YOLOv8 and YOLOv4 and detailing the advantages that led us to choose YOLOv8 for this study.

Accepted: We thank the reviewer for identifying the inconsistency in the chemical formula formatting. We have carefully reviewed and corrected the chemical formulas

Accepted: We appreciate the feedback on the flowchart's presentation. We have revised the flowchart

Accepted: Thank you for pointing out the issues with the confusion matrix. We have regenerated this figure to significantly improve its quality, ensuring the font is larger, clearer, and the image resolution is higher.

Accepted: We thank the reviewer for this insightful comment regarding the false positives for the "AS" class. We acknowledge that our initial explanation was insufficient. We have now expanded the discussion in the relevant section to provide a more in-depth analysis.

Accepted: Thank you for the feedback on the Conclusion. We agree that it needed refinement for conciseness and impact. We have substantially revised and shortened the Conclusion section

Reviewer 1

Thank you very much for your valuable comments

Reviewer 3

Thank you very much for your valuable comments

Reviewer 2 Report

Comments and Suggestions for Authors

View letter

  1. Keywords: biofertilizers seem to be unrelated to the manuscript's theme.
  2. Line 106  Where does the uniqueness or advantage of IGLOO manifest in terms of phosphorus characterization compared to other visual systems?
  3. Line 118  What are the differences between YOLOv8 and the previously mentioned in Introduction YOLOv4, and what are its advantages?
  4. Line 155-156  Note the case of chemical formulas.
  5. 4  The flowchart is fuzzy, and the overall layout can be optimized.
  6. 6, 7, and 8  The font of the confusion matrix is small and blurry; what does this matrix result represent?
  7. 10  The explanation for the insignificant reduction in false positives of the "AS" class after 500 training epochs is not sufficiently in-depth and requires a deeper exploration of the reasons, combining the principles of model training and the characteristics of the data.
  8. Fig.6-10 can be merged into one large figure for easier comparative analysis. Additionally, is Fig. 6-10 a simulated or real figure? It has a significant difference in shape compared to Fig. 3 and
  9. The conclusion is too long without a core point.

Author Response

(The authors gave the same response as above.)

Reviewer 3 Report

Comments and Suggestions for Authors

Comments to the authors

GENERAL COMMENTS
The manuscript titled: “IGLOO: Machine vision system for the determination of solubilization index in phosphate solubilizing bacteria”- Manuscript ID: microorganisms-3545106, after incorporating the reviewers' suggestions, is of significantly higher scientific merit. The authors of the manuscript addressed all the suggestions and made changes at the right level. 
Thus, I recommend this manuscript for publication in the "Microorganisms".

Author Response

(The authors gave the same response as above.)
